# 3D-Printed Overlay Template for Diagnosis and Planning Complete Arch Implant Prostheses

**DOI:** 10.3390/healthcare11081062

**Published:** 2023-04-07

**Authors:** Francisco X. Azpiazu-Flores, Damian J. Lee, Carlos A. Jurado, Hamid Nurrohman

**Affiliations:** 1Department of Restorative Dentistry, Gerald Niznick College of Dentistry, University of Manitoba, Winnipeg, MB R3E 0W3, Canada; francisco.azpiazuflores@umanitoba.ca; 2Director Advanced Prosthodontics Residency Program, College of Dentistry, The Ohio State University, Columbus, OH 43210, USA; lee.6221@osu.edu; 3Department of Prosthodontics, College of Dentistry and Dental Clinics, The University of Iowa, Iowa City, IA 52242, USA; carlos-jurado@uiowa.edu; 4Missouri School of Dentistry & Oral Health, A.T. Still University, Kirksville, MO 63501, USA

**Keywords:** overlay, implants, full mouth rehabilitation

## Abstract

Dental implants are a reliable alternative to treating edentulism. In clinical situations where the dentition has been severely affected by partial edentulism, advanced wear, or periodontal disease, establishing important occlusal elements such as the occlusal plane, incisal guidance, and esthetics can be hard to visualize at the diagnostic stage. Contemporary data-acquisition technologies such as 3D scanners and CAD/CAM systems permit the precise manufacture of highly complex devices applicable to any stage of restorative treatment. The present clinical report presents an alternative technique for evaluating the projected artificial tooth relationships, vertical dimension, and occlusal plane in patients with severely weakened dentition by using a 3D-printed overlay template.

## 1. Introduction

Dental implants are a powerful resource to rehabilitate partially and completely edentulous patients, which can efficiently improve esthetics, function, and quality of life [1,2,3,4,5,6,7,8]. A prosthetically-driven implant therapy has been advocated to guarantee accurate implant positions and definitive prostheses [4,5,6]. Traditionally, artificial teeth arranged following specific extraoral and intraoral anatomical landmarks were tried intraorally to establish the prosthetic contours that would determine the subsequent implant placement [9,10]. However, when the occlusal relationships are compromised by teeth migration, excessive wear, or chronic periodontal disease [11]; establishing and assessing important elements such as the vertical dimension, anterior teeth position, tooth and gingival display, smile width, occlusal plane, and lip support becomes a challenge [6,8,12,13]. Flangeless trial prostheses have been used to evaluate these factors in edentulous patients [6], and other methods such as intraoral mock-ups using bis-GMA acrylic resin have been used for partially edentulous patients. Regardless of the method, establishing these elements at the planning stage is a complicated labor when the intraoral conditions are unfavorable.

Research has demonstrated that complete arch implant-supported reconstructions restore the quality of life to a point equal to that of dentate patients [14,15], provide superior masticatory function than traditional removable prostheses [16], and have high survival rates [17]. A retrospective study evaluated 7-year clinical and 5-year radiographic outcomes for complete arch reconstruction supported by four dental implants and concluded that complete mouth reconstructions with a reduced number of implants were a feasible treatment modality that provided clinical and radiographic survival rates of 94.5% [17]. Furthermore, a recent retrospective cohort study evaluated the clinical outcomes of full-arch implant prostheses supported by two axial and two tilted implants followed for 12 to 15 years. In this study, a total of 692 dental implants were placed in 72 maxillae and 101 mandibles; a cumulative survival rate of 97.51% for the maxilla and 96.91% for the mandible after 15 years was reported, and the authors concluded that complete arch implant prostheses should be considered a viable treatment for the maxilla and mandible [18]. Factors such as improvements in implant surfaces [19], surgical planning protocols [5], and manufacturing technologies have contributed to these elevated survival rates [20,21,22].

Contemporary computer-aided design and computer-aided manufacturing (CAD/CAM) systems permit manufacturing objects with complex geometries accurately; for this reason, these technologies have been used in different fields such as engineering, medicine, and dentistry for decades [20,21,23]. Digital imaging and data acquisition technologies such as cone-beam computerized tomography (CBCT) and three-dimensional (3D) scanning are commonly used in combination to create the reference file that will be imported into the CAD software, and depending on how the desired object is fabricated, the CAM component can be additive or subtractive [21]. Additive manufacturing involves the creation of an object in a layer-by-layer fashion using a photopolymerizable polymer [22], while subtractive manufacturing involves the creation of the object using computer-controlled cutting tools [21]. Surgical templates [5], printed casts, customized impression trays [22], interim prostheses, prosthetic prototypes, and definitive prostheses can be manufactured accurately with these technologies [21]. Additionally, depending on the ingenuity of the dentist, custom-made appliances can be created with these technologies to manage complex clinical situations. The purpose of the present clinical report was to present an alternative approach for evaluating important prosthetic elements such as the projected artificial tooth relationships, vertical dimension, and occlusal plane on a patient with severely weakened terminal dentition by using a 3D-printed overlay template.

## 2. Materials and Methods

A 58-year-old patient arrived at the prosthodontic clinic at Ohio State University seeking comprehensive restorative treatment. During the initial examination, the patient expressed to be taking β-blockers and selective serotonin reuptake inhibitors (SSRI) to control his high blood pressure and anxiety. During the extraoral examination, facial features suggestive of a reduced vertical dimension of occlusion, including thinning of the lips, deepened labiomental and nasolabial grooves, and mild angular cheilitis (Figure 1a–c), were noticed.

The intraoral examination revealed partially edentulous maxillary and mandibular arches with an excessive vertical overlap and severe vertical and horizontal residual ridge deficiencies (Figure 2a). Multiple root tips, extensive wear facets, and restorations with secondary decay were observed in the remaining dentition (Figure 2b,c). The initial radiographic analysis corroborated the clinical findings and revealed severely resorbed posterior mandibular edentulous ridges, bilateral pneumatization of the maxillary sinuses, and bilateral antegonial notching (Figure 2d).

After discussing the clinical and radiographic findings, multiple treatment options involving removable and fixed tooth-supported restorations were presented to the patient. At this stage, the patient expressed not being interested in preserving his teeth since he was aware of the compromised condition of his dentition and was exhausted by the repeated failures of previous restorative attempts. Additionally, he expressed interest in improving his smile and preferred a definitive treatment capable of restoring the harmony between his dental and facial features. After considering these factors, a comprehensive treatment involving dental implants and complete arch implant-supported prostheses comprised of zirconia overlays supported by milled titanium bars was presented. The limitations, advantages, and possible complications of this treatment were discussed, and the treatment was accepted by the patient.

In the second appointment, maxillary and mandibular preliminary impressions were made with irreversible hydrocolloid (Geltrate; Dentsply Sirona North America, York, PA, USA), and centric relation at the desired vertical dimension was recoded with a bite leaf gauge (Bite Leaf Gauge; Patterson Dental, Saint Paul, MN, USA) and polyvinylsiloxane (PVS) bite registration material (Blu-Mousse; Parkell Inc., Edgewood, NY, USA). A facebow record and extraoral and intraoral photographs were also taken at this stage. Subsequently, diagnostic casts were fabricated and digitized using a benchtop 3D scanner (E3 Scanner; 3Shape A/S, Copenhagen, Denmark). The 3D models of the diagnostic casts generated were saved in Standard Tessellation Language (STL) format. Subsequently, a digital scan of the buccal surface of digital diagnostic casts with the bite registration was performed. The STL file of the buccal bite registration was used to align the maxillary and digital casts at the vertical dimension in a centric relation established clinically. This was performed using the point-based gluing option in the align tool of an open-source Standard Tessellation Language (STL) editor software (MeshLab; STI-CNR, Rome, Italy) widely used in dental research [24,25,26] (Figure 3a). Additionally, using another open-source 3D modeling software (MeshMixer 3.3.; Autodesk Inc., San Rafael, CA, USA), digital teeth arrangements (Figure 3b), blocked-out digital casts, and simulated gingiva were created for the subsequent design of maxillary and mandibular overlay templates (Figure 4, Figure 5, Figure 6, Figure 7 and Figure 8).

The overlay templates were manufactured using a stereolithographic (SLA) 3D printer (Form2; FormLabs, Somerville, MA, USA) with tooth-colored photopolymerizable resin (Denture Teeth B1 V1; FormLabs, Somerville, MA, USA). After completing post-curing, the 3D-printed templates were adjusted in the articulator to confirm the vertical dimension and occlusal plane orientation. Subsequently, these were tried intraorally (Figure 9), and afterwards the esthetics, phonetics, vertical dimension, position of the upper lip during maximum smile, and centric relation were evaluated clinically (Figure 10a–c).

Subsequently, the STL files of the overlay templates and digital casts were imported onto an implant planning software (SIMPLANT PRO 18.5; Dentsply Sirona North America, York, PA, USA). Using the digital file of the overlay templates as a reference, six standard-diameter dental implants (Tapered Screw Vent 3.7 × 11.5 and 3.7 × 10 mm; Zimmer Biomet, Parsippany, NJ, USA) and four standard diameter dental implants (Tapered Screw Vent 3.7 × 11.5 mm; Zimmer Biomet, Parsippany, NJ, USA) were planned anterior to the maxillary sinuses and in the mandibular anterior region between the mental foramina (Figure 11).

Subsequently, computer-generated bone reduction guides and bone-supported surgical templates were used to guide bone reduction and implant placement (Bone-Supported SIMPLANT Guide; Dentsply Sirona North America, York, PA, USA). At the time of the surgery, the implant placement occurred uneventfully, and maxillary and mandibular complete dentures were delivered and relined with tissue conditioner (COE-COMFORT; GC America, Alsip, IL, USA).

Four months later, at the uncovering stage, it was noticed that the most anterior left maxillary dental implant was mobile and ended up being removed uneventfully. After 3 weeks of healing, tapered abutments (Angled Tapered Abutment System; Zimmer Biomet, Parsippany, NJ, USA) were installed, and open-tray impression posts were splinted with low-shrinkage polymethylmethacrylate (PMMA) (Pattern Resin LS; GC America, Alsip, IL, USA) for definitive impressions with PVS (Aquasil Ultra Monophase; Dentsply Sirona North America, York, PA, USA). Subsequently, definitive casts were manufactured using type IV dental stone (Fuji Rock IMP; GC America, Alsip, IL, USA), artificial tooth arrangements were fabricated, and they were tried intraorally. After the occlusal relationships were deemed adequate, the artificial tooth arrangements were sent to the dental laboratory to be used as a reference for the fabrication of complete arch implant-supported prostheses consisting of zirconia overlays supported by titanium bars (AccuFrame 360; Cagenix Inc., Memphis, TN, USA). To guarantee satisfactory occlusion and esthetics, the lab was requested to send the titanium bars with 3D-printed overlay prototypes first to verify the passivity of the bar and refine the occlusion in the 3D-printed prototype. After the passivity of the titanium bars was verified, the anterior guidance, esthetics, phonetics, and occlusion were adjusted (Figure 12a). Additionally, a custom incisal guide table was fabricated with polymethylmethacrylate (Pattern resin LS; GC America, Alsip, IL, USA) to guarantee the reproduction of the desired occlusion in the definitive zirconia overlays (Figure 12b).

The definitive maxillary and mandibular complete implant-supported arches were tried intraorally, and the accurate reproduction of the desired occlusal relationships was confirmed (Figure 12c). At this stage, minimal adjustments were performed using 12µ metallic Shimstock film (Arti-Fol BK 28; Bausch BK, Nashua, NH) and fine-grit high-speed diamond burs (DOS1F FG; Brasseler, Savannah, GA, USA) followed by zirconia polishers (K0239 Dialite Zr Intra-Oral Polishing System; Brasseler, Savannah, GA, USA). Additionally, the complete seating of the prostheses was confirmed radiographically (Figure 12d), and the esthetics, phonetics, and restitution of facial support were reassessed and deemed satisfactory (Figure 12e).

Finally, maintenance instructions were provided to the patient, and an oral hygiene regime consisting of dental prophylaxis every 6 months and home maintenance with interproximal brushes (G.U.M. ProxaBrush; SUNSTAR Corp., Schaumburg, IL, USA) was established. At the time of delivery and the 1-week and 1-month recall appointments, the patient expressed satisfaction with the esthetics and function provided by the prostheses.

## 3. Results

CAD/CAM technologies simplify the management of complex complete arch rehabilitations by allowing the design and manufacture of custom-made appliances for specific clinical situations. In the present clinical report, the 3D-printed overlay templates were designed using open-source 3D modeling software and demonstrated to be a feasible alternative for the diagnostic and treatment planning of complete arch rehabilitations. Some of the advantages of this technique include: permitting the evaluation of important esthetic and functional elements prior to the surgical procedures; the design STL files can be imported into implant planning software to visualize the relationship of projected prosthetic contours with underlying anatomical structures; permitting the circumvention of undesirable events such as tooth fracture or getting material locked into severe teeth or soft tissue undercuts; and simplifying the diagnostic stage, thus streamlining the progression of the treatment. At the end of the treatment, the patient was satisfied with the esthetic and functional qualities provided by the definitive prostheses, expressing significant improvements in his self-esteem and quality of life.

## 4. Discussion

The present clinical report presents the successful use of a 3D-printed overlay for the treatment planning and rehabilitation of a patient with terminal dentition with dental implants. Traditionally, artificial teeth arrangements or intraoral mock-ups using composite resin have been used to evaluate the planned prosthetic contours, maxillomandibular relationships, vertical dimension, and esthetics before dental implants are placed for complete-arch implant-supported prostheses [4]. Although these approaches have been used successfully, both require caution when the dentition is weakened by extensive caries, wear, or advanced periodontal disease. As seen in the present clinical report, the removable 3D-printed overlay templates allowed for the avoidance of clinical complications such as tooth fracture and soft tissue damage related to mock-up material getting locked onto weakened teeth or pronounced soft tissue undercuts. This was possible since the 3D modeling software used allowed blocking out problematic undercuts digitally and permitted manufacturing an appliance that seated passively on the weakened dentition. Blocking out the cast can be a laborious and time-consuming procedure when multiple undercuts are present. This procedure involves multiple steps that demand careful attention and are prone to human error and material distortion. This source of error can be circumvented when digital workflows are used since multiple steps can be automated and the desired object can be designed without a physical working cast [22].

Three-dimensional printed overlay templates have been described in the literature for smaller restorative procedures in the esthetic zone and for treatment planning removable prostheses for completely edentulous patients. A recent clinical report described the use of a diagnostic wax-up and overlay template used to successfully improve the gingival tissue architecture of maxillary incisors using electrosurgery [27]. This approach allowed for effective communication of the prosthetically planned gingival architecture with the surgeon performing the procedure. A technique describing how to fabricate a 3D-printed overlay template to aid the placement of two implants in an edentulous mandible has been presented in the literature [28], and 3D-printed overlay tooth reduction guides have been advocated to minimize unnecessary reduction of flare-out or rotated teeth during veneer preparations [29,30,31].

Multiple research articles have demonstrated the versatility and the vast applications of CAD/CAM technologies in restorative dentistry. It has been reported that the accuracy of contemporary 3D printers can range from 61 to 92 µm with a production tolerance of less than 1 mm in the X, Y, and Z axes [22]. Manufacturing dental appliances with the details, thickness, and accuracy of the devices described in the previous articles and the one in this clinical report would have been extremely difficult to achieve with traditional methods such as compression, pouring, or injection molding. Traditional PMMA resins used for compression molding have a volumetric shrinkage of approximately 7%, and other materials such as composite resin can shrink from 1% to 4% with a polymerization shrinkage proportional to the volume of material applied [23]. Usually, analog manufacturing involves different materials to confine the prosthetic design to be molded [23]. The outstanding accuracy and precision of contemporary additive manufacturing systems permit creating the final appliance with dimensions very close to the digital design with minimal intermediate steps and minimal material usage [21]. For those reasons, 3D printing has become an attractive alternative to other CAD/CAM manufacturing technologies such as subtractive manufacturing [21]. Subtractive manufacturing, also known as milling, has been widely used in dentistry for the fabrication of tooth and implant-supported restorations, with zirconia being one of the most frequently used materials to manufacture dental prostheses. Research suggests that zirconia restorations, both tooth-borne and implant-supported, have elevated success rates, with one article reporting no significant differences in terms of complications between bruxers and non-bruxers [32]. Additionally, when designed with porcelain limited to their gingival portion, complete-arch implant-supported prostheses made of zirconia have demonstrated survival rates above 99% [33]. As a result, multiple clinical reports have been written depicting the use of this polycrystalline ceramic to manage complex clinical situations that demand a combination of strength and biocompatibility; patients with craniofacial [34], autoimmune conditions [35], and other conditions have been successfully rehabilitated with one-piece complete arch implant-supported prostheses, and case-control studies suggest the adequacy of this material to rehabilitate bruxers with extensive occlusal wear and reduced vertical dimension [36]. For these reasons, in the present clinical report, a monolithic zirconia occlusal overlay with minimal pink porcelain was used since this prosthetic design would permit predictably rehabilitating the esthetics, function, and vertical dimension of the patient while minimizing the risk of complications.

The present clinical report presents significant limitations related to its limited follow-up time, the lack of quantitative analysis of the outcomes, and the fact that it only describes a single patient and its clinical situation. Two major limiting factors of the 3D-printed overlay technique are that modern additive manufacturing technologies are required for the fabrication of the appliance, limiting its offer to practices or dental laboratories that have this equipment, and that its manufacture involves expenses related to the photopolymer used, thus resulting in additional laboratory and clinical fees. Another limitation of the present clinical report is that other more contemporary registration techniques, such as digital impressions with implant scan bodies, were not performed, therefore making the present clinical report a combination of digital and analog procedures and not a fully digital workflow, which is progressively becoming the norm in contemporary full-arch implant prosthodontics. However, regardless of its limitations, the 3D-printed overlay template is an efficient alternative to establishing and assessing important treatment elements in patients with weakened teeth, large soft tissue undercuts, and compromised maxillomandibular relationships, which can be manufactured in open-source 3D modeling software. Finally, this clinical report presented a step-by-step description of the functions used to block out digital casts and create templates by means of subtraction. In the present clinical situation, the large undercuts present in the buccal surface could not be blocked completely since the alveolar gingiva of the patient was slightly anterior to the teeth when the undercuts were determined with the “select visible tool”. To overcome this situation, a second block-out was performed, which permitted eliminating these effectively. This was possible given the open nature of the software, which provided more freedom to the operator; this “freedom of design” represents an advantage since the different steps of the presented workflow can be advantageously adapted to other challenging clinical situations at the discretion of the clinician.

## 5. Conclusions

The current clinical report presents the comprehensive complete-arch rehabilitation of a patient with terminal dentition, in which a 3D-printed overlay template was used to assess and test the desired vertical dimension, occlusal plane, and incisal edge position. Contemporary data acquisition and CAD/CAM technologies allow for the design and fabrication of accurate dental devices that simplify the execution of complex complete-mouth rehabilitations. 3D-printed overlay templates are a viable alternative for the non-invasive intraoral evaluation of the projected prosthetic design in patients with terminal dentition, which can be predictably designed using open-source 3D modeling software.

## Figures and Tables

**Figure 1 healthcare-11-01062-f001:**
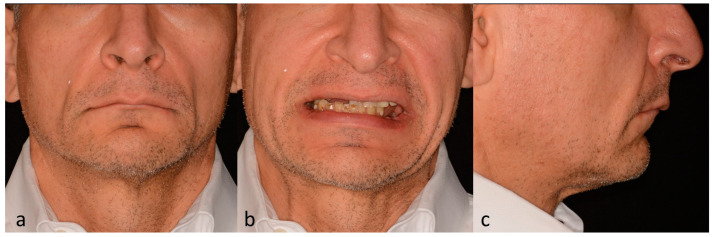
Initial situation. (**a**) Closed mouth. (**b**) Smile frontal view. (**c**) Lateral view with a closed mouth.

**Figure 2 healthcare-11-01062-f002:**
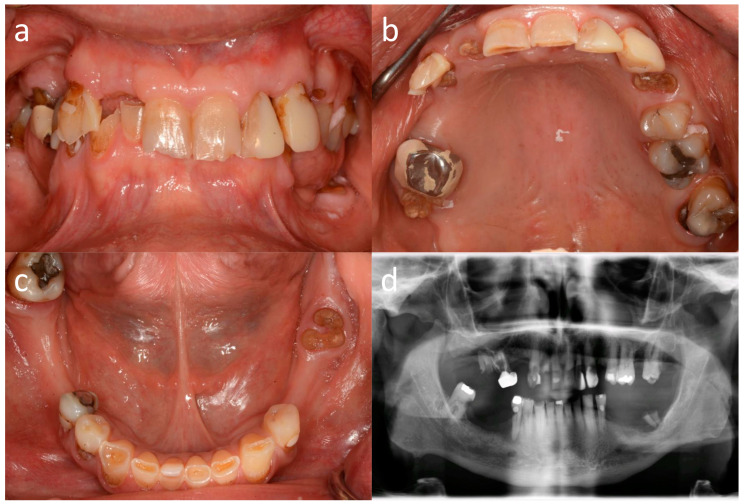
Initial situation. (**a**) Frontal view. (**b**) Occlusal view of the maxillary arch. (**c**) Occlusal view of the mandibular arch. (**d**) Panoramic radiograph.

**Figure 3 healthcare-11-01062-f003:**
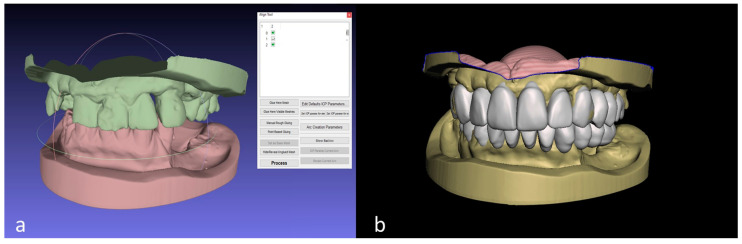
Digital maxillary and mandibular casts and digital tooth arrangements. (**a**) Digital casts are related in a centric relation at the desired vertical dimension of occlusion, determined intraorally using leaf gauge and bite registration material. Scans related to the bite scan using multiple surface points as references were created using the point-based gluing option in the STL editor software (MeshLab; STI-CNR) (name of the files removed for privacy reasons). (**b**) Digital tooth arrangements performed in open-source 3D modeling software (MeshMixer 3.3.; Autodesk Inc., San Rafael, CA, USA). The Christian Brenes Tooth library was imported into the modeling software and oriented using the transform tool; markings previously scribed in the buccal surface of the cast and extraoral photographs were used as references for the incisal edge position and establishing the occlusal plane.

**Figure 4 healthcare-11-01062-f004:**
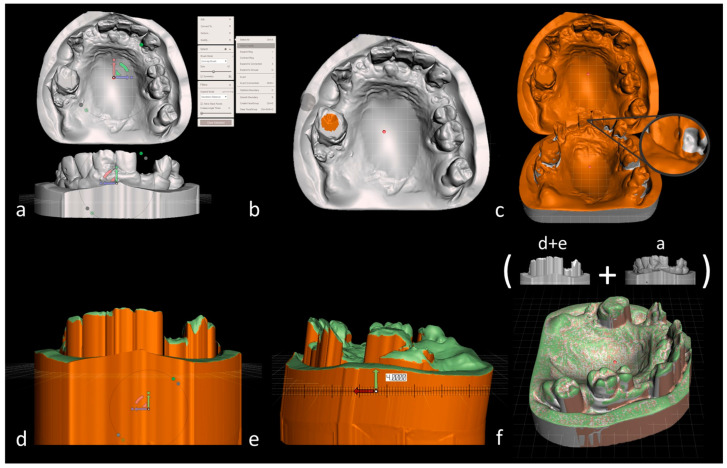
Workflow used to identify the undercuts and blocking out the maxillary digital cast. (**a**) Digital maxillary cast with its center aligned in the world origin of the 3D modeling software. Note the Y-axis perpendicular to the occlusal plane, anteroposterior X-axis, and mediolateral Z-axis. (**b**) Determination of the path of insertion and undercuts using the operator’s view of the digital cast as reference using the “select visible “function. (**c**) Undercuts requiring blockout (grey). Note the careful inspection of the cast to select small pinpoint areas to prevent problems with the blockout procedure. (**d**) First blockout (vertical) performed by displacing the undercut areas in the -Y direction (orange) using the transform tool. (**e**) Second blockout performed in a duplicate of the digital cast. Note displacement in the X+ direction to eliminate large anterior soft tissue undercuts. (**f**) Merging of first and second blocked-out casts (**d**+**e**) with digital maxillary cast (**a**). At this stage, the internal irregularities of both blocked-out models were filled with the “make solid” function in the modeling software before combining them with the original maxillary digital cast to create the final blocked-out maxillary digital cast.

**Figure 5 healthcare-11-01062-f005:**
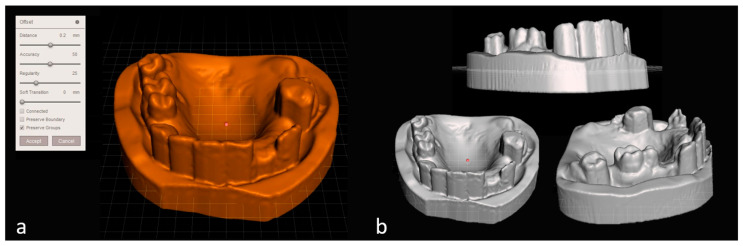
Finalizing the blocked-out maxillary digital cast. (**a**) A 0.2mm offset applied to the blocked-out cast to ensure passive fitting of the overlay. At this stage, the reference blocked-out cast must be deleted by inverting the selection. (**b**) A finalized maxillary blocked-out cast.

**Figure 6 healthcare-11-01062-f006:**
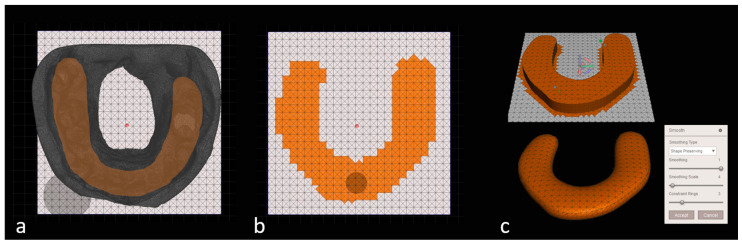
Designing the gingival portion of the overlay template. (**a**) A plane appended to the world origin of the modeling software. Using the unselected maxillary digital cast as reference, an approximation of the arch shape was selected on the appended plane. (**b**) Arch shape transferred to the appended plane. (**c**) Displacement of the selected area in the Y+ direction to create a gingival “horseshoe”. Note the softened contours achieved using the “Smooth” function.

**Figure 7 healthcare-11-01062-f007:**
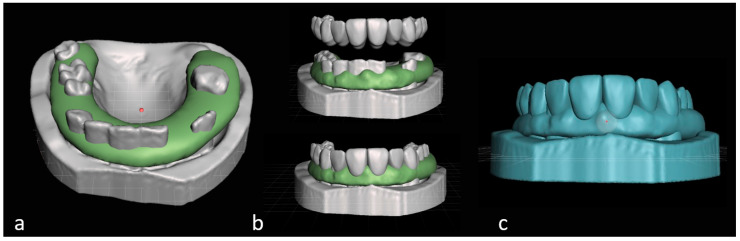
Modeling of the gingival horseshoe and digital teeth. (**a**) Gingival horseshoe positioned on the blocked-out model. (**b**) Digital tooth arrangements previously set in the models at the desired vertical dimension of occlusion. (**c**) Digital design of maxillary overlay after minor modifications.

**Figure 8 healthcare-11-01062-f008:**
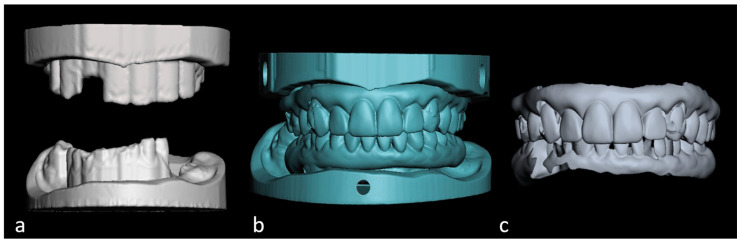
Finalized digital blocked-out casts and digital overlay templates. (**a**) Maxillary and mandibular blocked-out casts. (**b**) Maxillary and mandibular final overlay designs. (**c**) Overlay templates after subtraction of finalized blocked-out digital casts using the “Boolean difference” function.

**Figure 9 healthcare-11-01062-f009:**
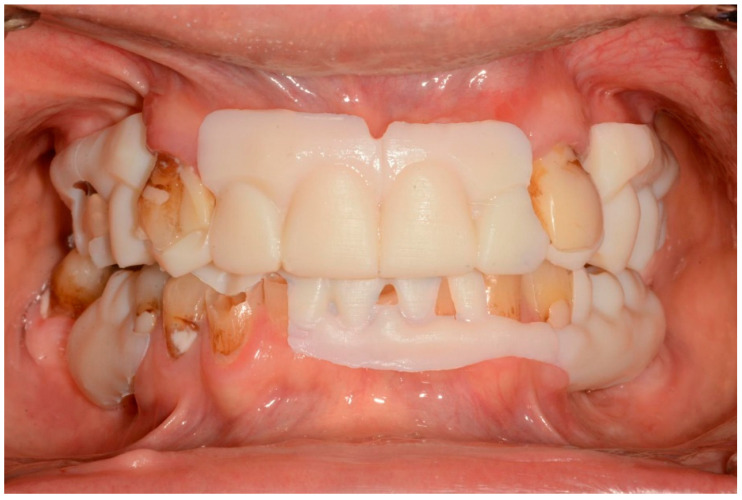
Intraoral evaluation of the 3D-printed overlay templates. Note satisfactory re-establishment of the vertical dimension of occlusion, occlusal plane, and anterior teeth relationships with these non-invasive diagnostic appliances.

**Figure 10 healthcare-11-01062-f010:**
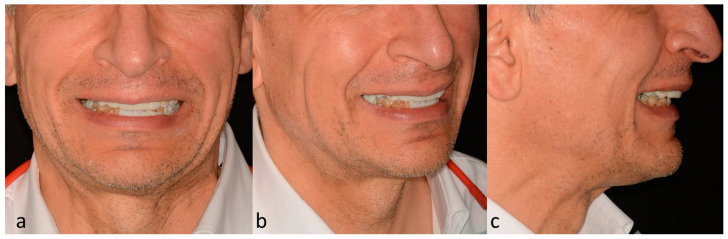
Extraoral evaluation of the 3D-printed overlays. (**a**) Smile frontal view. (**b**) Smile lateral view. (**c**) Smile profile view.

**Figure 11 healthcare-11-01062-f011:**
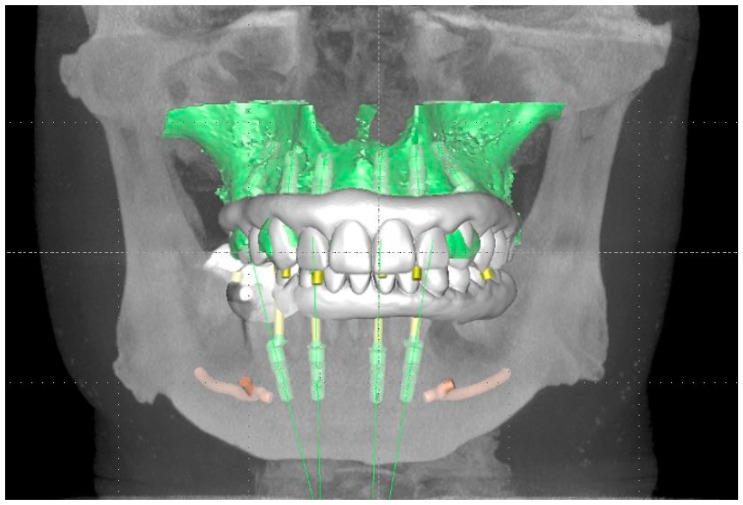
Frontal view of digital planning using the contours of the digital designs of the 3D-printed overlay templates as reference.

**Figure 12 healthcare-11-01062-f012:**
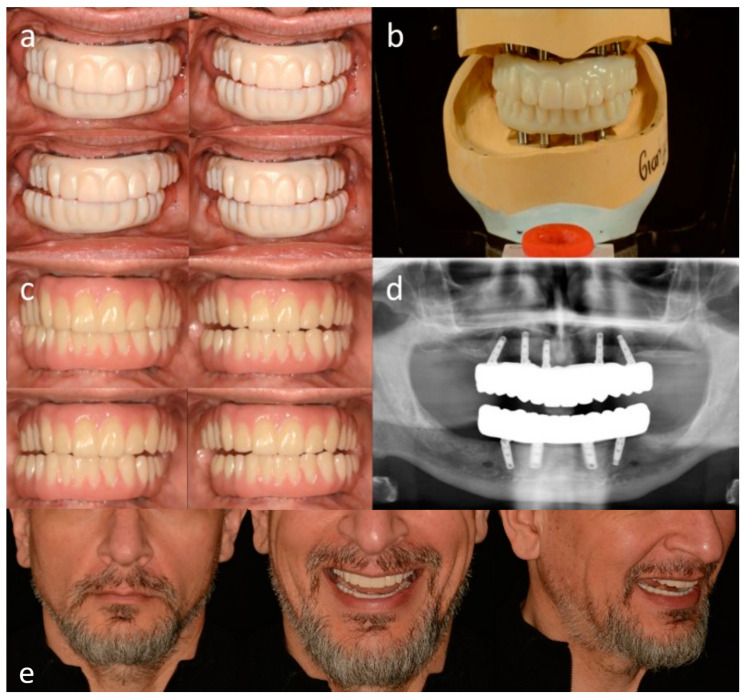
3D-printed prototypes and definitive prostheses. (**a**) Intraoral try-in of bars and 3Dprinted overlays. (**b**) Articulated 3D-printed overlays and titanium bars with a custom incisal guide table (**c**) Intraoral photograph of definitive prostheses (**d**) The final panoramic radiograph. (**e**) Extraoral photographs.

## Data Availability

Not applicable.

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
