# Peer review of "3D-Printed Overlay Template for Diagnosis and Planning Complete Arch Implant Prostheses"

_healthcare, 2023, doi:10.3390/healthcare11081062_

Round 1
Reviewer 1 Report
1, Why didn't the authors take conservative treatment of tooth. It should never be pulling teeth without careful consideration. It is better to have more teeth, as much as possible. Because, even if that is the case, pulpless teeth are able to function as sensory organ. Is this treatment method only suitable for edentulous patients? Please add more in Method and Discussion.
2, Since zirconia overlays is too hard than traditional materials. The authors should describe about "How about para-functional force?" Please add more in Discussion.
Author Response
1, Why didn't the authors take conservative treatment of tooth. It should never be pulling teeth without careful consideration. It is better to have more teeth, as much as possible. Because, even if that is the case, pulpless teeth are able to function as sensory organ. Is this treatment method only suitable for edentulous patients? Please add more in Method and Discussion.
ANSWER: Dear reviewer, we appreciate your evaluation and comments for improving our manuscript. We like to maintain teeth whenever they are in good conditions and provide proper function. Unfortunately, the presented patient had terminal dentition with several missing posterior teeth, fractured teeth, severe wear, teeth with secondary caries and occlusion was collapsed as you can see in figure 2. This patient was carefully screen at the Ohio State University and patient and clinicians opted to have a full mouth reconstruction with implant therapy. Discussion section was extended with more information.
2, Since zirconia overlays is too hard than traditional materials. The authors should describe about "How about para-functional force?" Please add more in Discussion.
ANSWER: Thanks for your evaluation. We did not provide zirconia overlays, we provided an printed overlay resin template and the final implant restorations were made out of zirconia. Zirconia implant restorations have shown to be very successful with long-term positive outcomes.
A recent retrospective study evaluated the survival rate of zirconia implant restorations for bruxer versus non-bruxer patients for 58 months, and it concluded with no significant differences in the overall survival and success rate of zirconia restorations in bruxer versus non-bruxer patients.
Heller H, Sreter D, Arieli A, Beitlitum I, Pilo R, Levartovsky S. Survival and Success Rates of Monolithic Zirconia Restorations Supported by Teeth and Implants in Bruxer versus Non-Bruxer Patients: A Retrospective Study. Materials (Basel). 2022 Jan 22;15(3):833. doi: 10.3390/ma15030833. PMID: 35160777; PMCID: PMC8836879.

Reviewer 2 Report
Digital technology is supposed to make "dental life" easier for the treatment team and more predictable for the patient. Many workflows are still not ready making digital approaches somewhat questionable.
I would like to comment as follows:
Why have all teeth been extracted? Based on Fig. 2 there should have been a chance to retain at least some teeth and restore with a removable restoration on teeth and implants
Why was a conventional impression made for the final restoration? Scanning implants with scan bodies in my opinion currently constitutes the most reliable digital procedure
As the "overlay" was not 1:1 transferred to the final prostheses I do not see why this technique would have any benefit - a regular mock-up using a vacuum-formed matrix as retainer would have served the same purpose
Author Response
REVIEWER #2
Digital technology is supposed to make "dental life" easier for the treatment team and more predictable for the patient. Many workflows are still not ready making digital approaches somewhat questionable.
I would like to comment as follows:
Why have all teeth been extracted? Based on Fig. 2 there should have been a chance to retain at least some teeth and restore with a removable restoration on teeth and implants
ANSWER: Dear reviewer, thank you for your kind observations.
Certainly, some of the teeth could be retained, however, their unfavorable occlusal relationships (extrusion of the mandibular teeth, and dentoalveolar compensation with extensive lingual wear on the maxillary anterior teeth), and the fact that multiple restorative treatments would be required to restore the teeth to establish an adequate occlusal plane was unattractive to the patient and for that reason he decided to get complete arch implant supported prostheses.
A few sentences were added to the materials and methods section detailing this (Lines 101-111).
Why was a conventional impression made for the final restoration? Scanning implants with scan bodies in my opinion currently constitutes the most reliable digital procedure
ANSWER: Dear reviewer, you are totally correct. Intraoral scanning is progressively becoming the norm for complete arch implant impressions. However, with absolute sincerity, we didn’t have the scan bodies for that implant system available at that time, therefore, very unfortunately, intraoral scanning was not an option for this treatment. A paragraph was added at the end of the discussion expressing this as a limitation. (Lines 316-334)
Thank you for your observations.
As the "overlay" was not 1:1 transferred to the final prostheses I do not see why this technique would have any benefit - a regular mock-up using a vacuum-formed matrix as retainer would have served the same purpose
ANSWER: Dear reviewer, thank you for your kind observation.
Certainly, the 3D printed template was not transferred to the final restorations, however, their contours permitted the planning and subsequent placement of the implants in favorable positions that permitted an uncomplicated and successful rehabilitation. One of the main concerns with doing a regular mock with a vacuum-formed template and bysacrylic resin was the weakened condition of the patient’s dentition and the presence of large soft tissue undercuts and dental cavities onto which bisacrylic composite could get locked fracturing the teeth during removal and/or causing soft tissue lacerations. However, we agree that for most clinical situations a vacuum-formed template is an excellent technique to establish important prosthodontic references at the diagnostic stage. Concurring with your comment a sentence in the discussion states that a limitation of the clinical report is that it only presents the management of a single patient and clinical situation and therefore is an “alternative approach” to other traditional techniques such a making a mockup using a vacuum-formed template.

Reviewer 3 Report
1. It is necessary to present the protocol and parameters by which the alignment in occlusion of digital models is made in the Meshlab application, as well as the demonstration that this alignment is similar to commercial CAD applications. 2. It is not explained how the recording with the facebow was digitally transferred in Meshmixer 3. It is not explained how the virtual teeth library was realized and used in the particular case of the report 4. The way in which the gingival tissues were modeled in Meshmixer should be explained 5.
It is absolutely necessary that the way in which the block out of the model was achieved with the help of the "Transform" tool is explained step by step, as it involves multiple transformations that cannot be verified / reproduced by the readers
Author Response
REVIEWER #3
- It is necessary to present the protocol and parameters by which the alignment in occlusion of digital models is made in the Meshlab application, as well as the demonstration that this alignment is similar to commercial CAD applications.
ANSWER:
Dear reviewer, a description of the options used for the alignment in MeshLab was added to the manuscript (For details refer to the Legend of Fig.3 and Lines 122-128). MeshLab is a 3D analysis software commonly used to evaluate the trueness and has a very accurate alignment tool; this software has been used for dental research purposes to evaluate the accuracy and trueness of dental casts. I included some references to the document of studies that used this software for dental research purposes (Please see line 124)
Also, please note that due to patient confidentiality reasons the name of the files and some files were not included.
Thank you for your kind observations.
- It is not explained how the recording with the facebow was digitally transferred in Meshmixer
ANSWER: Dear reviewer, the facebow recording was not transferred to MeshMixer, only the static relationship of the maxillary and mandibular casts at the desired vertical dimension. The template designed in MeshMixer was used for diagnostic and planning purposes only to establish incisal edge position, depth of placement of the implants, approximate a-p spread. A conventional facebow record was performed later in the treatment for the definitive prostheses.
Thank you for your kind observations.
- It is not explained how the virtual teeth library was realized and used in the particular case of the report
ANSWER: Dear reviewer, the details regarding how the digital teeth arrangements were used and a brief description of the creator of the library were included in the manuscript (Lines 135-143.)
- The way in which the gingival tissues were modeled in Meshmixer should be explained 5.
It is absolutely necessary that the way in which the block out of the model was achieved with the help of the "Transform" tool is explained step by step, as it involves multiple transformations that cannot be verified / reproduced by the readers
ANSWER: Dear reviewer, thank you for your kind observations. As per your request, multiple images and legends were added describing the workflow used to block out the digital casts and create the templates. Please refer to Figures 4.1-4.6 and their legends for details.
Kindest regards

Round 2
Reviewer 1 Report
I can not find in Discussion about para-functional force.
The authors should describe about "How about para-functional force?" Please add more in Discussion about following sentences :The final implant restorations were made out of zirconia. Zirconia implant restorations have shown to be very successful with long-term positive outcomes.
This article concluded with no significant differences in the overall survival and success rate of zirconia restorations in bruxer versus non-bruxer patients. Please add this article in references.
Heller H, Sreter D, Arieli A, Beitlitum I, Pilo R, Levartovsky S. Survival and Success Rates of Monolithic Zirconia Restorations Supported by Teeth and Implants in Bruxer versus Non-Bruxer Patients: A Retrospective Study. Materials (Basel). 2022 Jan 22;15(3):833. doi: 10.3390/ma15030833. PMID: 35160777; PMCID: PMC8836879.
Author Response
Reviewer: I can not find in Discussion about para-functional force.
ANSWER:
Dear reviewer, we apologize for the missing information. The updated version of the manuscript is addressing it to the best of our possibilities. We also included a paragraph in the discussion where the advantages of using zirconia to treat patients with parafunctional habits are discussed.
Reviewer: The authors should describe "How about para-functional force?" Please add more in Discussion about following sentences :The final implant restorations were made out of zirconia. Zirconia implant restorations have shown to be very successful with long-term positive outcomes.
ANSWER:
Dear reviewer, thank you for your recommendation. We added a few sentences in the discussion portion where the benefits of zirconia as a restorative material for patients with parafunctional habits are discussed. Also, we added the literature you recommended to the manuscript.
Thank you!
Reviewer: This article concluded with no significant differences in the overall survival and success rate of zirconia restorations in bruxer versus non-bruxer patients. Please add this article in references.
Heller H, Sreter D, Arieli A, Beitlitum I, Pilo R, Levartovsky S. Survival and Success Rates of Monolithic Zirconia Restorations Supported by Teeth and Implants in Bruxer versus Non-Bruxer Patients: A Retrospective Study. Materials (Basel). 2022 Jan 22;15(3):833. doi: 10.3390/ma15030833. PMID: 35160777; PMCID: PMC8836879.
ANSWER:
Dear reviewer, thank you for your recommendation; as per your request, the article was added, and it was mentioned in the discussion section of the manuscript. I appreciate your help.
Reviewer 2 Report
I am sorry to say that this may be good didactic material but does not suit a scientific journal
Author Response
Dear Reviewer:
Thank you for your review. We set a high standard and are sorry we failed to meet your expectations.
Sincerely,
Authors
Reviewer 3 Report
my observations were answered in a satisfactory manner
Author Response
Dear reviewer:
Thank you for your kind words! We appreciate your comment.
Sincerely,
Authors